# Compounding impact of severe weather events fuels marine heatwave in the coastal ocean

B. Dzwonkowski [1,2 ✉], J. Coogan[1,2,5], S. Fournier[3], G. Lockridge[2], K. Park [4] & T. Lee [3]

Exposure to extreme events is a major concern in coastal regions where growing human populations and stressed natural ecosystems are at significant risk to such phenomena. However, the complex sequence of processes that transform an event from notable to extreme can be challenging to identify and hence, limit forecast abilities. Here, we show an extreme heat content event (i.e., a marine heatwave) in coastal waters of the northern Gulf of Mexico resulted from compounding effects of a tropical storm followed by an atmospheric heatwave. This newly identified process of generating extreme ocean temperatures occurred prior to landfall of Hurricane Michael during October of 2018 and, as critical contributor to storm intensity, likely contributed to the subsequent extreme hurricane. This pattern of compounding processes will also exacerbate other environmental problems in temperature-sensitive ecosystems (e.g., coral bleaching, hypoxia) and is expected to have expanding impacts under global warming predictions.

[1] University of South Alabama, Mobile, AL 36688, USA. [2] Dauphin Island Sea Lab, Dauphin Island, AL 36528, USA. [3] Jet Propulsion Laboratory, California Institute of Technology, Pasadena, CA 91109, USA. [4] Texas A&M University at Galveston, Galveston, TX 77554, USA. [5] Present address: Woods Hole Oceanographic Institution, Woods Hole, MA 02543, USA. ✉email: briandz@disl.org

The resiliency of natural and human systems is often tested by extreme events. Such events alone can be devastating and when coupled with existing environmental stresses and/or low frequency trends, they can serve as critical change points[1,2]. As a result, advancing the understanding of extreme events is fundamental to risk and vulnerability assessments that support management decision making and policy development. This need for improved understanding is becoming especially important in coastal regions where growing human populations, infrastructure, and stressed natural ecosystems present significant exposure to catastrophic impacts from extreme events[3,4]. A primary extreme event of concern for coastal systems throughout lower and mid-latitude regions is tropical cyclones. There are numerous examples of the devastating impacts these storms have had on coastal communities[4], yet advances in intensity forecasts have been limited[5]. This is particularly problematic for storms making landfall where accurate intensity prediction is critical to human safety[5].

Forecasting storm intensity, as with other extreme events, is challenging due to contributions from multiple drivers (e.g., storm track, wind shear, dry air entrainment, and air–sea interaction) with their amplification potential dictated by complex causal chains[6]. More specifically, the compounding processes that intensify storms and generate extreme conditions in coastal regions may differ from those in the open ocean due to the presence of shallow shelf bathymetry and coastline[7,8]. These constraints can influence the way sea surface temperature (SST), the critical thermodynamic interface for ocean–storm interaction, evolves and thus modifies storm intensity[9]. Several studies have shown that the presence of stratification prior to storm arrival can significantly reduce the intensity of impending storms, via ahead-of-eye SST cooling[7,8,10,11]. Conversely, other studies have suggested or shown that anomalously warm conditions on shelves have contributed to the intensification of hurricanes through landfall[6,12–14].

While the previous studies showed connections between the shelf thermal structure and coastal storm intensification, the threats associated with the amplification of storms making landfall necessitate a broader understanding of processes by which shelf heat content can be pushed to extreme levels. This critical gap in understanding is in large part due to the lack of observational data during extreme events, which by their definition are rare. Such data could put events in historical context and/or determine the antecedent conditions and process(es) that generate such events[4]. Despite these challenges, a long-term coastal observing system in the northern Gulf of Mexico provides a unique perspective on shelf thermal conditions before, during, and after Hurricane Michael, an extreme storm event in October 2018. Through the extensive observational records at this mooring site as well as an idealized one-dimensional model, this study demonstrates that the shelf heat content, prior to Hurricane Michael making landfall, was at an extreme state (i.e., experiencing a marine heatwave) set up by a series of compounding atmospheric processes: a shelf mixing event by tropical storm (TS) Gordon followed by an atmospheric heatwave. While this study emphasizes the connection between shelf heat content and storm intensification, compound events that push the shelf to an extreme state (i.e., a full water column marine heatwave) represent a potentially important means by which other temperature-sensitive coastal issues can be amplified into extreme states (e.g., coral bleaching, hypoxia), demonstrating the broad importance of this series of compounding processes.

The development and amplification of hurricanes in the Gulf of Mexico are a common phenomenon during summer and fall seasons (i.e., the Atlantic hurricane season between June 1 and November 30) due to areas of very high heat content derived

from the Loop Current and its associated eddies[15,16]. Thus, the intensification of Hurricane Michael (October 7–10, 2018, Fig. 1) fits a common pattern for this region; however, this event did stand out as an extreme on several metrics[17]. The strength of the storm alone, reaching a rare category 5 designation, demonstrates the extreme nature of the event. More striking is the setup under which this strengthening occurred as the storm developed late in the Atlantic Hurricane Season and intensified throughout its transit across the continental shelf. Both of these factors are typically associated with weaker or a weakening of storm events[18]. Furthermore, this storm was particularly hazardous because its intensification was rapid and was consistently underpredicted by forecasts throughout its life[17]. Consequently, Hurricane Michael, the strongest storm on record to make landfall on the Florida Panhandle, resulted in 16 fatalities and $25 billion in damage in the United States[17]. Initial investigations of this storm highlight several factors that may have contributed to intensification including favorable atmospheric and oceanic conditions[17,19–21]. However, the amplification through landfall suggests the thermal structure and associated heat content on the shelf in the Mississippi Bight may have played a contributing role to the evolution of this extreme storm event.

## Results

**Shelf heat content evolution.** The temperature data at site CP provide a unique view of shelf heat content and its potential to have contributed to the intensification of Hurricane Michael (Fig. 2a). Prior to landfall, the thermal conditions on the shelf were exceptionally warm with the depth-average temperatures, a proxy for heat content[9], generally above 29 °C during the latter half of September into October. The heat content only dropped by ~0.5 °C over the 3–4 days when Michael transited across the shelf, leaving the conditions well above the 26 °C threshold typically considered conducive for storm intensification[9]. The pre-storm period is particularly striking when put in context with historical data at site CP. The long-term (2005–2018) mean in late September is around 27 °C (standard deviation of ~1 °C), much closer to the lower temperature limit associated with

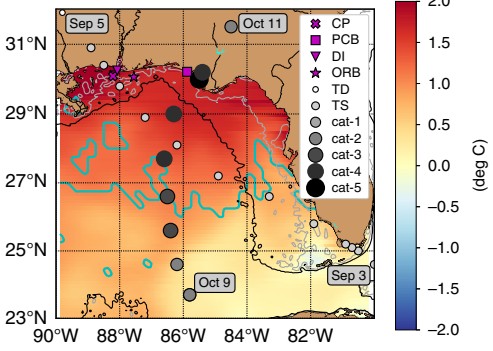

**Fig. 1 Map of the study region.** Map of the eastern Gulf of Mexico showing 10-day average OISST anomalies relative to a long-term climatological mean (1981–2019) between September 27 and October 6 with the region between the cyan contour and land showing the primary area that experienced a marine heatwave within that time period. Closed cyan contours outside or inside this area show pockets with or without marine heatwaves, respectively. Also shown are storm tracks for TS Gordon (lighter gray circles) and Hurricane Michael (darker gray circles) with reference dates and storm categories as well as locations of in situ data stations across the Mississippi Bight: site CP (CP, **X**), DPIA (DI, ★), Orange Beach Buoy (ORB, ▲), and Panama City Beach Fishing Pier (PCB, ■). The gray and black contours are the 20 and 100 m isobaths, respectively.

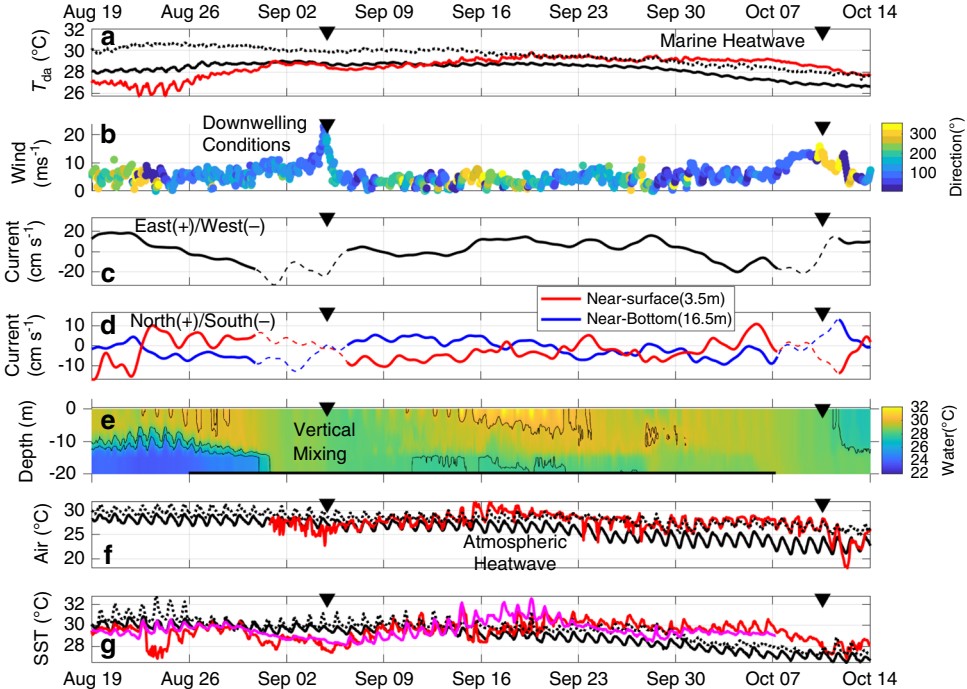

**Fig. 2 Time series of observations.** Time series during the fall of 2018 for **a** depth-average temperature at site CP (red), a proxy for shelf heat content, **b** wind speed and direction (coloration) at ORB with 0°/360° being a north wind, **c** along-shelf depth-average currents at site CP, **d** across-shelf currents at site CP, **e** vertical temperature structure at site CP, **f** air temperature at DI (red), and **g** SST at PCB (red) and site CP (magenta), with the inverse triangles at the top indicating the landfall times of TS Gordon (9/4/2018) and Hurricane Michael (10/10/2018). Also shown in (**a**), (**f**), and (**g**) are the long-term mean (black line) and the 90th percentile threshold (dotted line) based on 13, 32, and 5–9 years of data, respectively. In (**b**), the blue (yellow) colors indicate downwelling (upwelling) conditions. In (**c**) and (**d**), the thin dashed lines indicate the periods associated with potentially low-quality data due to movement of the AWAC instrument. In (**e**), the black contours indicate isotherms (26, 28, and 30 °C) and horizontal black line at the bottom indicates the time period of the model simulations (cases A1–A3 in Table 1).

intensification (~26 °C). By comparison, the 2018 depth-average temperature was well above these conditions (at or above the 90th percentile), making it the warmest year observed in the in situ time series at site CP and exceeding any other year by 0.5–1 °C for this late September/early October time period.

To arrive at this high level of thermal energy (i.e., a full water column marine heatwave), the evolution of the depth-average temperature experienced two critical periods of increasing heat content. The first increase from ~26.5 to ~29 °C (August 25–September 2) was driven by a downwelling event caused by southeast winds in late August (Fig. 2b–d), associated with the approach and arrival of TS Gordon. The onshore Ekman transport driven by the easterly component of the southeast winds brought warm surface waters across the shelf which are then forced down at the coast, thereby initiating the warming in the subsurface. The subsequent increase of wind speed enhanced the vertical mixing of the warmer surface and cooler subsurface waters, which further homogenized the temperature in the water column. The event resulted in a dramatic change in the temperature structure with the highly stratified shelf becoming vertically uniform (Fig. 2e). As the storm passed over the region, site CP lost ~0.75 °C in the depth-average temperature consistent with some thermal dynamic heat loss expected from tropical storms, i.e., heat transferred from the ocean to the atmosphere[9]. Despite the heat loss, the vertical mixing of surface water increased the bottom temperature by ~4 °C relative to pre-storm conditions.

The well-mixed conditions lasted several days, at which point the second warming period began (September 6–22) and was associated with relatively mild wind conditions (wind

magnitudes $<{\sim}5\,\mathrm{m\,s^{-1}}$) and excessively warm atmospheric conditions (air temperature $>{\sim}28$ °C) (Fig. 2b, f). For context, most of the mid to late September 2018 air temperatures were consistently above the 90th percentile threshold from climatological September air temperatures at marine stations throughout the region (Fig. 2f and Supplementary Fig. 1). This atmospheric heatwave produced a warming event of longer duration (relative to TS Gordon), reheating the upper water column resulting in depth-average temperatures >29 °C and restratifying the thermal structure with the SSTs exceeding 32 °C at times (Fig. 2e, g). Interestingly, there was a mid-water column warming in late September (~28th–29th, Fig. 2e). From the available data it is difficult to determine the cause of this event, however, the warming may have been generated by advection or density compensation. The overall effect of these processes resulted in a water column that maintained extreme heat content well into early October when coastal waters usually begin to experience rapid cooling. Given the thermal conditions at site CP, two natural questions arise in relation to Hurricane Michael. First, is the coupling of events (i.e., a storm mixing event followed by an atmospheric heatwave) critical to the observed excessively high heat content on the shelf in early October? Second, are the conditions at this shelf site (i.e., site CP) representative of the broader Mississippi Bight where Hurricane Michael intensified (Fig. 1)?

**Compounding impacts on heat content.** To address these questions, a relatively simple one-dimensional (vertical) model, similar to the one in ref. [22], was used to provide a first-order

**Table 1 Design of model runs.**

| ID | Initial conditions | Surface heat flux | Comments |
|---|---|---|---|
| Three runs to test the model (August 25–October 7) | | | |
| Case A1 | 8/25 condition[a] | NARR and S&B[b] | Salinity specified with data |
| Case A2 | 8/25 condition | NARR and TOGA (with observed SST)[c] | Salinity specified with data |
| Case A3 | 8/25 condition | NARR and TOGA (with modeled SST)[c] | Salinity specified with data |
| Three runs to examine the importance of the storm mixing event (September 7–October 7) | | | |
| Case B1 | 8/25 condition | NARR and S&B | |
| Case B2 | 8/25 condition (mixed)[a] | NARR and S&B | |
| Case B3 | 8/25 condition (mixed) | NARR and S&B | With a generic storm heat loss[d] |
| Eight runs to examine the depth dependency of the compound impact[e] (September 7–October 7) | | | |
| Case C1a, C2a, C3a, C4a | Stratified | NARR and S&B | |
| Case C1b, C2b, C3b, C4b | Vertically mixed | NARR and S&B | |

[a]The observed thermal profile at 23:00 on August 25, 2018 and its vertical average (black solid and black dashed lines in Fig. 4, respectively): The initial depth-average temperatures are 27.3 °C for cases A's and B's, and 28.8, 27.5, 26.1, and 23.4 °C for C1, C2, C3, and C4, respectively.
[b]Net outward radiation estimated using the bulk formulations in ref. [22].
[c]Latent and sensible heat flux estimated using the TOGA-COARE algorithms in ref. [35] forced with observed or modeled SST.
[d]With a generic storm heat loss based on an open ocean category 3 storm following ref. [9].
[e]Idealized thermal profiles with water depths of 20, 30, 40, and 60 m (cases C1–C4: see Fig. 4) with vertically uniform salinity.

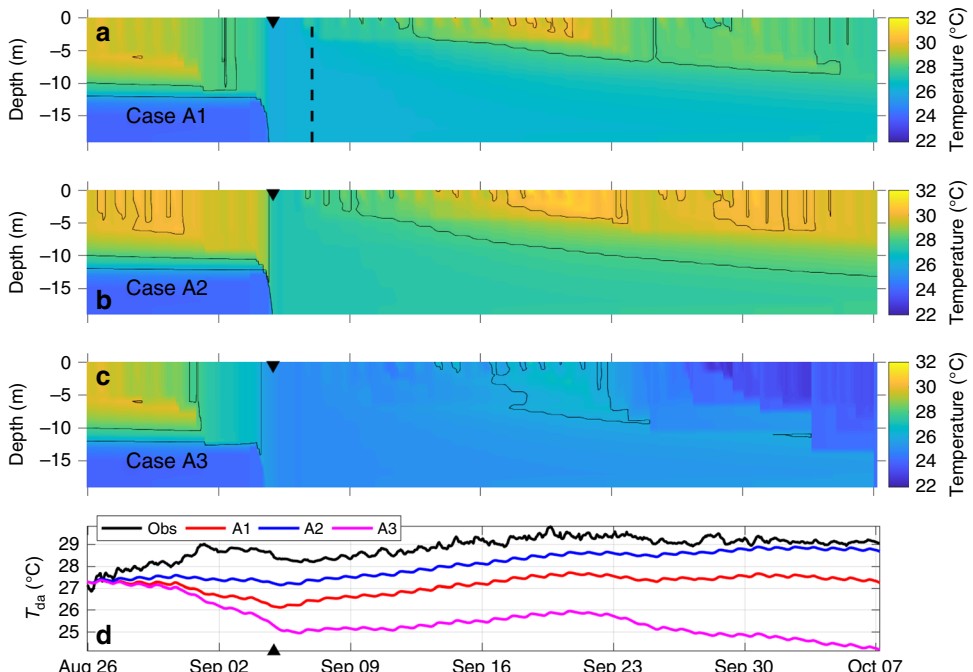

**Fig. 3 Times series of the model results for cases A1–A3.** The modeled temperature structure from cases **a** A1, **b** A2, and **c** A3 as well as **d** the comparison between the observed (Obs) and modeled depth-average temperature, with the filled triangles indicating the landfall time of TS Gordon (9/4/2018). The three model cases differ in the parameterization of the heat flux terms and their use of observed or modeled SST with details provided in the "Methods" section and Table 1. Accounting for the cold bias associated with response to TS Gordon, case A1 in (**d**) best captured the depth-average temperature response after the impact of TS Gordon with $r = 0.89$. The case A1 parameterization was used in the subsequent model experiments (cases B's and C's in Table 1) with the black vertical dashed line in (**a**) marking the post TS Gordon period (9/7/2018) when the subsequent model runs began.

understanding of the processes impacting the thermal structure on the shelf as well as the relative importance of the coupled events. The one-dimensional model simulates the time evolution of the vertical profile of the temperature by applying surface heat fluxes as well as mixing energy determined from bulk formulas for wind and tidal currents at the surface and bottom, respectively. The design of different model runs is summarized in Table 1 with details in the "Methods" section. In short, three simple model scenarios were conducted to determine (1) the best available surface heat flux parameterization; (2) the impact of storm mixing; and (3) the effects of varying shelf depths.

The initial model runs (cases A1–A3 in Table 1) were designed to determine how well a one-dimensional model may capture the main features of the observed thermal structure with three different heat flux parameterization methods (see "Methods" section). For these runs, a subset of the data between August 25 and October 7 (horizontal black line in Fig. 2e) was selected to examine the model performance over the period around the two main events hypothesized to contribute to the late October extreme heat content. The results of cases A1–A3 captured several important aspects of the observed thermal structure (Fig. 3) despite the inherently three-dimensional nature of coastal heat

budgets[23]. All three forcing approaches demonstrated: (1) the complete mixing of the water column by TS Gordon resulting in the warming of the lower portion of the water column, and (2) the post-Gordon thermal re-stratification associated with the reheating of the upper ocean.

It is clear that this one-dimensional approach fails to reproduce the initial warming in late August and the timing of the homogenization of the water column associated with TS Gordon (Figs. 2e and 3a). These features in the temperature structure are characteristics of downwelling events[24], which highlights the established importance of three-dimensional processes associated with hurricane responses in the coastal ocean[7,8]. This model limitation imparts a cold bias in temperature outputs at the start of the post TS Gordon period with the depth-average temperatures in the different A cases being between ~1 and 3 °C cooler than the observations (~September 7, Fig. 3d). While case A2 had SST and depth-average temperature closer to the observed conditions, the heat flux parameterizations used in case A1 was selected for the subsequent model experiments for two reasons. First, the latent and sensible heat fluxes in case A2 were derived with the observed SST which represents a dependency on a priori information nudging the model outputs toward the observations that they are being compared to. Furthermore, this same parameterization of the heat flux when using the modeled SST (case A3), i.e., without the observationally derived heat fluxes, did much worse relative to case A1 (Fig. 3c, d). Second and more importantly, of the three modeled cases, case A1 showed the best representation of the observed post-Gordon depth-average temperature variations, after accounting for the cold bias associated with the passage of TS Gordon (Fig. 3d). The relatively high correlation between case A1 and observations ($r = 0.89$) indicates that the post-storm thermal structure of the water column was primarily a one-dimensional balance driven by surface heat fluxes and vertical mixing. Thus, this simple one-dimensional model (case A1) was used in the subsequent model experiments to further examine the thermal structure with and without compounding processes.

Given the reasonable results produced by the one-dimensional model after TS Gordon, we conducted experiments focused on the specific time period between September 7 and October 7 to assess the impact of the mixing on the evolution of the thermal structure, particularly in terms of the depth-average temperature, prior to the atmospheric heatwave. The first set of numerical experiments (cases B1–B3 in Table 1) were conducted using the thermal structure just prior to TS Gordon (either stratified as observed or artificially mixed: black lines in Fig. 4) as initial conditions and allowing them to evolve based on forcing conditions during the model run. The results were notably different with the stratified water column remaining stratified while the mixed case restratified beginning in early September but remained much more weakly stratified due to the warm thermal conditions at depth (Fig. 5a, b). This difference in stratification had an impact on the uptake/loss of heat across the air–sea interface with the mixed cases (cases B2 and B3) having ~1 °C higher depth-average temperature than the stratified case (Fig. 5c). Given that the observed depth-average temperature was nearly 2 °C above the long-term mean in early October, this ~1 °C increase from the mixing event coupled with a subsequent atmospheric heatwave accounts for nearly 50% of the observed deviation from the long-term mean state (Fig. 2a) and represents a temperature change large enough to significantly impact storm intensity[25–27]. Importantly, removing this added 1 °C effect from the observed depth-average temperature in early October would place the depth-average temperature well below the 90th percentile threshold associated with marine heatwaves (Fig. 2a). Thus, the compounding processes observed in September of 2018, adding 1 °C depth-average temperature to

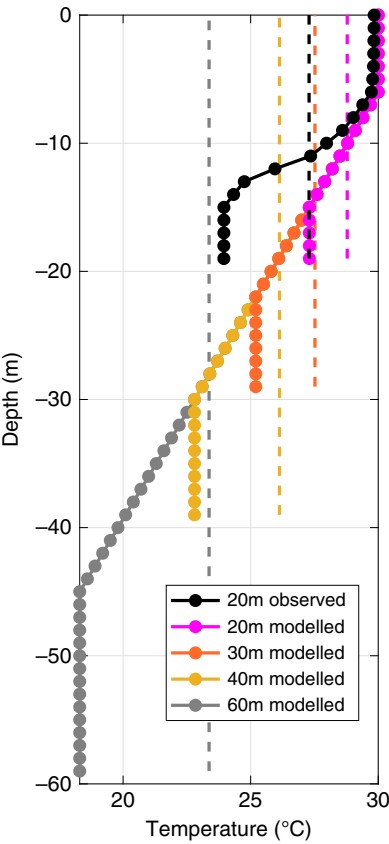

**Fig. 4 Initial temperature structure for the different model cases.** The observed profile (black solid line) and its vertical average (black dashed line) are from site CP at 23:00 on August 25, 2018, and used as the initial thermal structure for the model runs with '8/25 condition' and '8/25 condition (mixed)' in Table 1, respectively (cases A1–A3 and B1–B3). The colored solid and dashed lines indicate the idealized stratified (Cases C1a, C2a, C3a, and C4a) and mixed (Cases C1b, C2b, C3b, and C4b) initial temperature conditions, respectively, for the four additional model scenarios with different depths (Table 1). Note that the vertical dash line indicates the depth-average temperature associated with a given stratified case, projected over the full water column to represent 'mixed' temperature structure for that scenario (e.g., Cases C1a and C1b).

the water column, made what would have been an otherwise above average event into an extreme event (above the 90th percentile threshold, Fig. 2a).

Additional numerical experiments using idealized thermal profiles more typical of the northern Gulf of Mexico over a range of depths (cases C1–C4 in Table 1 and Fig. 4) indicate a depth-dependent pattern with the relative impact increasing with increasing depth (Fig. 5d). In these model experiments, the focus was on the differences in the evolution of the depth-average temperatures between the initially stratified and mixed conditions at a given water column depth. The idealized 20-m water column (case C1) had less thermal stratification than the observation-based case and hence a smaller difference in the upper ocean temperatures between the mixed and stratified cases (Fig. 4). As a result, the stratified and mixed initial conditions (cases C1a and C1b, respectively) showed very little effect on the overall heat content of the water column, i.e., very little differences in the depth-average temperatures (Fig. 5d). This changed as deeper profiles allowed for larger differences in the upper ocean temperatures between the mixed and stratified cases. In the two deepest cases (i.e., 40 and 60 m), the event coupling provided differences >0.8 °C in the depth-average temperatures between

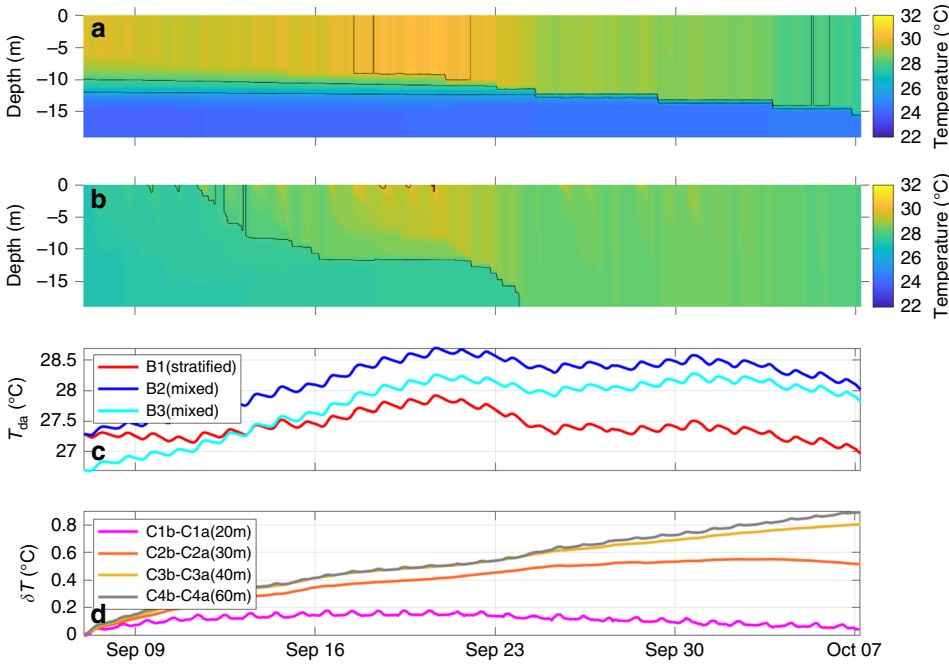

**Fig. 5 Results of model experiments that compare the temperature characteristics of the water column with and without a mixing event.** The water column temperature structures with the **a** stratified and **b** mixed initial conditions, i.e., without and with the storm mixing (cases B1 and B2 in Table 1, respectively); the black contours in (**a**) and (**b**) indicate isotherms (26, 28, and 30 °C). **c** The depth-average temperature from cases B1 (red, stratified), B2 (blue, mixed), and B3 (cyan, mixed with initial heat loss from a storm event); note that the mixed cases (B2 and B3) end nearly 1 °C warmer than the stratified case (B1). **d** The depth-average temperature difference between the mixed (cases C1b–C4b) and stratified (cases C1a–C4a) model experiments with idealized temperature profiles with water depths of 20 (magenta), 30 (orange), 40 (yellow), and 60 m (gray); see Table 1 for a summary description of different cases and Fig. 4 for the initial temperature profiles used. For all model experiments, the net outward radiation was estimated using the bulk formulations in ref. [22].

the mixed and stratified cases (Fig. 4d). In addition, these modeling scenarios suggest that the increasing heat content difference with water depth will eventually be mitigated by the depth-average temperature dropping to or below 26 °C (e.g., gray dashed line in Fig. 4), leading to conditions that would not be expected to favor hurricane intensification. While additional observational and modeling work is needed to fully understand relationships between mixing and reheating events on continental shelves, our findings do indicate that changes in depth and hydrographic structure will, not surprisingly, affect the extent to which compounding processes intensify the warming of the water column. Overall, the coupling of a mixing event followed by an atmospheric heatwave does enhance heat content of the water column relative to an atmospheric heatwave without a mixing event.

**Potential extent of the compound event**. While the model results indicate that the compounding impact of the sequential atmospheric events (i.e., TS Gordon and then the atmospheric heatwave) did contribute to the extreme state of heat content at site CP, its location is notably distant away from the track of Hurricane Michael (Fig. 1). There is some evidence suggesting that the processes observed in the western Mississippi Bight were similar to those in the eastern part of the basin where Hurricane Michael crossed the continental shelf. First, the shelf mixing associated with TS Gordon likely impacted the broader Mississippi Bight region as the structure of the system had a significant wind field. This is supported by satellite data that showed notable decreases in SST ($\Delta T$ of 1–2 °C) and increases in sea surface salinity ($\Delta S$ of 1.0–1.5), indicative of mixing across the shelf throughout the region (Fig. 6). The response to the atmospheric heatwave was also regional in nature as indicated by the

exceptionally warm satellite-derived SST anomalies across the region, consistent with a marine heatwave (Fig. 1). The only in situ data on the shelf in the eastern Mississippi Bight was SST data at site PCB. Similar to site CP, the water temperature was anomalously high relative to historical values, and more importantly the in situ SST data between site CP and PCB for 2018 followed very similar patterns including the warming prior to TS Gordon and the slower, longer duration of warming associated with the regional atmospheric heatwave (Fig. 2g). Thus, the extreme heat content observed at site CP was indicative of a regional marine heatwave in the Mississippi Bight that would have resulted in the intensification of Hurricane Michael.

## Discussion
Regardless of the exact contribution of shelf heat content on the intensification of Hurricane Michael, this study has identified a new pattern of compounding processes that can lead to extreme conditions in coastal oceans (Fig. 7). The observed extreme thermal conditions were set up by an initial mixing event during the passage of TS Gordon and intensified by a subsequent regional atmospheric heatwave. While the downwelling effect associated with TS Gordon was clearly important in influencing the heat content at this specific site, the storm generated mixing that impacted the broader shelf region (Fig. 6) and proved critical in two ways. First, the direct mixing of shelf water resulted in a vertical redistribution of the thermal properties, transferring the upper ocean heat down to deeper portions of the water column (Fig. 7b), which effectively removed all the cooler bottom water from the shelf.

Second, the reorganization of the thermal structure created a water column better able to absorb subsequent heat input, thus allowing a rapid re-warming of the upper ocean (Fig. 7c). This

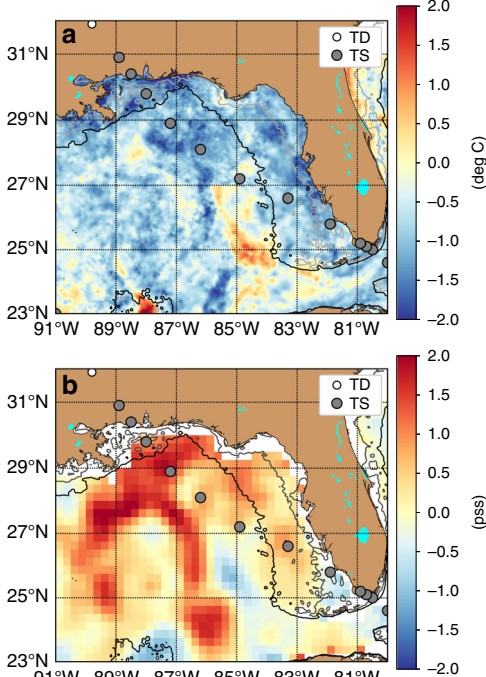

**Fig. 6 Surface temperature and salinity differences.** Surface ocean property differences in **a** temperature and **b** salinity before and after the passage of TS Gordon, with the circles showing the storm track of Gordon as tropical storm (TS) and depression (TD). Dictated by data availability, the temperature difference was determined using the conditions on August 28 and September 7, 2018, and the salinity difference using September 2 and September 9, 2018. The gray and black contours are 20- and 100-m isobaths.

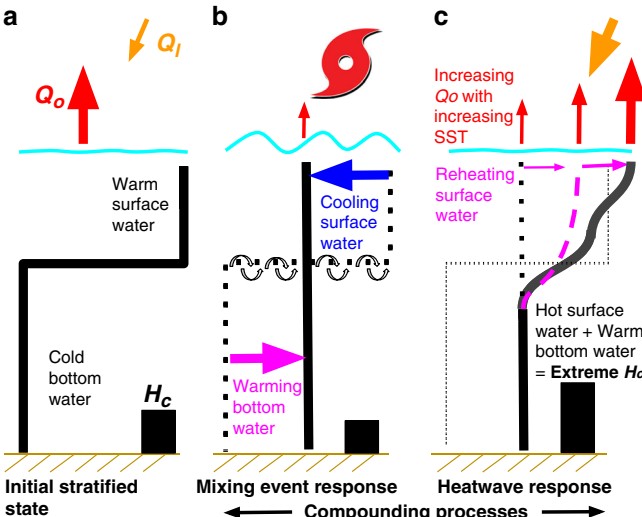

**Fig. 7 Conceptual diagram of the compound event.** Conceptual diagram of the compound event that resulted in extreme shelf heat content including the **a** initial stratified shelf, **b** mixing event response, and **c** atmospheric heatwave response, showing the water column temperature profile (black thick line), the water column heat content ($H_c$: thick black bar on the bottom), the incoming solar radiance ($Q_i$: yellow arrow), and the net heat loss ($Q_o$: red arrow). The dashed black line in (**b**) and (**c**) indicates the temperature structure from a previous phase of the sequence of events. The magenta dashed line in (**c**) indicates changing upper-layer stratification during the atmospheric heatwave. The changing arrows for $Q_o$ reflect the dependence of this term on SST.

was caused by the reduced SST post-mixing event which lowered the effectiveness of the primary heat loss terms (i.e., sensible and latent heat fluxes). The reheating of the upper ocean was further facilitated by initially weak stratification levels due to storm mixing which were conducive to a rapid redistribution of incoming solar radiation to deeper waters. Since solar insolation is primarily absorbed by the very near surface of the water column, low levels of stratification would allow this incoming heat to be more easily mixed to depth, limiting the rate of temperature increase at the sea surface and thus slowing the heat loss out of the upper ocean. Consequently, the compounding impacts of a mixing event followed by an atmospheric heatwave resulted in the highest late September/early October heat content on the shelf in the Mississippi Bight over a 13-year record.

An unclear aspect of the newly identified sequence of events is the duration of the bottom warming associated with the downward transfer of heat after a mixing event (Fig. 7b). This question directly relates to the residence time of bottom waters which, being dependent on shelf geometry and circulation, is likely to be highly variable among different shelf regions. The Mississippi Bight is generally a wide (~200 km) and gently sloping (depth to width ratio of ~0.001) shelf (with the exception of the centrally located Desoto Canyon region), which would favor long residence times. Furthermore, current velocity data from site CP after TS Gordon showed a series of upwelling and downwelling events producing limited net transport between September 7 and October 7 with a depth-average mean of ~3.3 cm s$^{-1}$ which translates to a transport length scale of ~85 km. While this information is from only a single location, it suggests the broader regional circulation was likely weak, favoring a longer retention of the warm bottom water.

The importance of heat content and thermal structure on hurricane intensity is well-established and much of the recent work on continental shelves has focused on intensity reduction through stratification breakdown and the resulting injection of colder bottom water into the upper water column as storms pass over these regions. Our findings highlight that this same destratification process, given the right conditions, can be a precursor for extreme heat content, and hence intensification of subsequent storms. The shelf-wide extent of the processes observed at the mooring site was consistent with the regional marine heatwave occurring across the Mississippi Bight and would have contributed to the observed intensification of Hurricane Michael as it transited the continental shelf. With near-real-time availability of many coastal in situ and satellite observations and an understanding of this type of compound event, prediction of potential storm intensification of landfall events could be enhanced. Thus, this study reinforces the urgent need to better represent coastal hydrographic conditions in hurricane prediction efforts.

Unfortunately, the understanding of extreme events is complicated by climate change, and current long-term climatic trends suggest a growing positive feedback between the processes involved in this compound event that could expand the impact of such extreme conditions. For example, tropical storms are expected to increase in strength and there is some evidence of a poleward expansion of their activity[28]. In addition, there is abundant evidence that terrestrial heatwaves will increase in frequency, duration, and intensity[29]. As a result of these shifts in the event characteristics, mixing events are likely to impact wider swaths of shelf areas and the subsequent reheating is likely to be more intense. Thus, the cumulative effect should amplify the impacts of this compound event pattern on shelf heat content under the current forecasts for a warmer climate.

While the statistical likelihood of this compound event is unclear now as well as in the future, the importance of

understanding extreme conditions lies in their disproportionately large impact on human and natural systems. As suggested by this study, extreme hurricanes, being linked to SST, are clearly one class of coastal hazard that will be influenced by such extreme heat content. However, this compounding amplification of heat content may be devastating to ecosystems as well, which are typically of significant societal and economic value. For example, temperature-sensitive benthic communities and habitats (e.g., coral reefs and hypoxia-prone shelves) already stressed by long-term warming trends and terrestrial inputs may be pushed beyond their resilience capacity by such extreme heat events[4,30]. While the effects of marine heatwaves on ecosystems have been demonstrated[31,32], the impacts of the observed extreme heat content events are difficult to quantify using conventional SST-based identification algorithms because of the significance of the water column temperature structure at depth. As such, this sequence of compounding processes and the resulting extreme conditions represent a 'black swan' event that a range of coastal interests should be considering in management and disaster response decisions.

## Methods

**In situ data and associated analysis.** To understand the atmospheric and oceanic conditions on the shelf before, during, and after Hurricane Michael, field and reanalysis data from various sources were used. Standard meteorological data, including air temperature, relative humidity/dew point temperature, and wind speed and direction from two NOAA National Data Buoy Center (NDBC) stations were used: 42012 offshore of Orange Beach, AL (ORB) and DPIA1 on Dauphin Island (DI) (Fig. 1). These stations were the closest measurements available to the main mooring site (CP), and both were typically similar in nature and have relatively long records (10 and 32 years at ORB and DI, respectively). Other key atmospheric variables, including incoming solar radiation and outgoing long-wave radiation, were obtained from the National Centers for Environmental Prediction (NCEP) North American Region Reanalysis (NARR) for the grid cell closet to site CP (https://www.esrl.noaa.gov/psd/data/gridded/data.narr.html). The NARR outputs are on 1/3° grid (~32-km resolution) and were interpolated from the 3-h outputs to hourly to match the NDBC data.

The hydrographic data were primarily derived from a long-term mooring station (site CP) on the 20 m isobath to the west-southwest of Mobile Bay (Fig. 1). The site provides a relatively long-term (13 years in 2005–2018) perspective of shelf thermal structure from a suite of instruments throughout the water column. The instrument suite has changed over time, but typically consists of bottom (~0.4 m above bottom, mab) and near-surface (~15.5 mab) CTD instruments, and 4–9 thermistors. Details of the mooring configurations and aspects of the processing can be found in various studies[33,34]. Importantly, the summer of 2018 featured an additional data stream from a CTD on a surface buoy (~100 m apart from site CP), providing data at ~19.5 mab (i.e., ~0.5 m below the surface). In order to provide a similar in situ SST measured in previous years, SST data from ORB (~1 m below the surface) were combined with those from site CP for available data (9 years in 2009–2017). These data are available at the NOAA NCEI (https://accession.nodc.noaa.gov/0203749) and Dauphin Island Sea Lab Alabama Real-time Coastal Observing System (https://arcos.disl.org/). Additional time series data for surface conditions were obtained from NOAA NDBC station PCBF1 (2005–2018) on the Panama City Beach Fishing Pier (PCB, Fig. 1), closer to the landfall of Hurricane Michael. A few relatively minor gaps were present in some of the time series data which were filled using either linear interpolation or through substitution of data from a nearby station. The resulting time series were used to determine ensemble properties (i.e., means, standard deviations, and maximum values) for the long-term records. One exception was PCB, which had significant gaps in the times series and resulted in ensemble properties determined using 5–9 years of data as availability allowed. These ensemble properties for in situ SST and air temperature were consistent with an overlapping 11-year time series from an inshore station in St. Andrews Bay (2008–2018), ~20 km east-southeast of PCB.

The heat content/flux calculations were carried out following typical procedures. For heat content, the depth-average temperature was used given the shelf focus of the study[9,12,13] and was calculated with water column temperature observations interpolated to a 1-m grid. While surface temperature is the means though which tropical cyclones interact with the ocean, this study focuses on depth-average temperature as averaging over the expected mixing depth of a storm has been suggested or shown be to the most relevant property for storm intensification[35–37], particularly over a shallow shelf[9,12]. The sensible and latent heat fluxes were calculated with the TOGA-COARE algorithms[35] similar to other studies in the region[34]. Conversion between relative humidity and dew point temperature was conducted following the standard algorithms (https://bmcnoldy.rsmas.miami.edu/Humidity.html)[38–40]. Extreme thermal events in the ocean and

atmosphere were determined following the methods in ref. [41]. In short, a marine heatwave is determined to be temperatures exceeding the 90th percentile for five or more days while an atmospheric heatwave required two or more days. As in ref. [41], the climatological means and 90th percentiles for the time series data were produced at each time step (hourly for site CP) which follows the recommendation in ref. [41] to use the longest, highest resolution data available.

Current velocity data throughout the water column were obtained from a Nortek Acoustic Doppler Waves And Current profiler (AWAC) at site CP. The data were processed following the procedure in ref. [42], where a 40-h low-pass Lanczos filter was used to highlight synoptic scale circulation patterns. Note that the periods around the storm events have potentially lower quality data due to movement of the instrument but the data appear to be physically consistent with the wind forcing. As such, the data were shown, but are indicated as questionable (Fig. 2c, d). These data are available at the Dauphin Island Sea Lab Data Management Center (https://www.disl.org/research/data-management-center) and NOAA NCEI (https://accession.nodc.noaa.gov/0211052).

**Satellite data and associated analysis.** For determining the regional sea surface conditions and delineating the impacts of TS Gordon on the shelf, several satellite-based data sets were used. SST data were obtained from the Jet Propulsion Laboratory (JPL) Multi-scale Ultra-high Resolution (MUR) SST product (https://podaac.jpl.nasa.gov/Multi-scale_Ultra-high_Resolution_MUR-SST) which is available from 2003 to the present. This is a 1-km resolution product that blends 1-km infrared sensor data and 25-km microwave sensor data. While this SST data set is relatively short in duration compared to other SST products, its high resolution is ideal for the coastal environment, the focus of this study. In addition, we used the recently released NOAA Optimum Interpolated SST (OISST) version 2.1 (https://www.ncdc.noaa.gov/oisst) for 1982–2019. Similar to MUR, this product blends infrared and microwave data, but also includes ship, buoy, and Argo float observations. While courser in resolution (0.25°), this data set is much longer than the MUR SST and has been commonly used in heatwave studies[41], making this work readily comparable to recent marine heatwave studies.

As with the mooring data, the methods of ref. [41] were followed to identify locations that experienced a marine heatwave within the 10-day period (September 27–October 6) prior to the arrival of Hurricane Michael in the Gulf of Mexico and identified as a full water column marine heatwave in the mooring data. The time period covers the dates when the depth-average temperature at site CP was consistently above the 90th percentile value until 3 days prior to the landfall of Hurricane Michael. The averaging period was ended 3 days prior to landfall to avoid including any storm impacts, which is similar to the way (see ref. [43]) in which a pre-storm SST value was obtained . While the MUR SST did appear to be generally consistent with the OISST data, the OISST data were used in the spatial analysis of the marine heatwave for the reasons mentioned above, i.e., duration and consistency with previous studies[31,44–46].

In addition, sea surface salinity (SSS) data were obtained from the NASA Soil Moisture Active Passive (SMAP) mission (https://podaac.jpl.nasa.gov/SMAP). The JPL SMAP SSS L3 product was used (8-day composite on a 1/4° grid, i.e., ~25-km resolution). As a result of this coarse spatial resolution, pixels by the coast are frequently contaminated by land and are excluded from the analysis. Note that salinity values presented are based on the practical salinity scale.

**Modeling setup and scenarios.** A simple one-dimensional (vertical) model, similar to the one in ref. [22], was used to explore the primary processes impacting the thermal structure of the water column at site CP. The model requires surface heat flux and mixing energy to simulate the evolving hydrographic structure of the water column. The mixing energies at the surface and bottom boundaries were determined using bulk formulas as functions of wind speed and tidal current, respectively[22]. The wind energy for mixing was determined from wind data at ORB, the closest oceanic wind measurement source to site CP. Since the Mississippi Bight is a microtidal environment, the tidal mixing was determined using a small fixed current value (12 cm s$^{-1}$). NARR outputs were used for solar radiation and other heat fluxes were estimated using two approaches: (1) NARR outputs for back radiation and sensible and latent heat estimated using the TOGA-COARE algorithms in ref. [35] and (2) net outward radiation estimated using the bulk formulations in ref. [22].

The first set of model runs (cases A1–A3 in Table 1) were conducted to determine how well a one-dimensional model may capture the main features of the observed thermal structure in response to the different heat fluxes parameterizations. A subset of the data (August 25–October 7) was selected to examine the period around the two main events hypothesized to contribute to the late October extreme heat content. After October 7, there was no longer surface data available at site CP. While this was still three days before landfall of Hurricane Michael, the applicability of the one-dimensional model is expected to breakdown as three-dimensional ocean processes became increasingly important on the shelf as Hurricane Michael approached. As such, October 7 served as a natural end point of the model simulations. The model runs were initialized with the observed temperature and salinity profile data (interpolated to a 1-m grid) on August 25. Salinity plays a significant role for the stratification at site CP[47], but the salinity data were very limited (only three vertical depths). With linearly interpolated salinity specified at each time step, only the temperature structure evolution was

modeled. The model was run for three cases: net outward radiation estimated using the bulk formulations in ref. [22] (case A1), and sensible and latent heat estimated using the TOGA-COARE algorithms in ref. [35] forced with observed (case A2) or modeled (case A3) SST. From the comparison between the modeled thermal structure and depth-average temperature time series to data (see the "Results" section), the heat flux parameterization in case A1 was selected as most appropriate and used for the subsequent model experiments.

A second set of model experiments (cases B1–B3 in Table 1) were designed to examine the compounding impacts of the storm mixing event and atmospheric heatwave. To exclude the three-dimensional nature of storm impacts by TS Gordon, this set of experiments was conducted from September 7 (after landfall of TS Gordon on September 4) until October 7. However, the initial conditions were based on the observed stratified temperature and salinity profiles on August 25 prior to TS Gordon so the evolution of the thermal structure of the water column just prior to TS Gordon could be evaluated with and without the impacts of a mixing event. As such, three scenarios were tested, differing only in the structure of the initial conditions. In case B1 the observed thermal (black solid line in Fig. 4) and salinity profiles were used as initial conditions. Because the temperature structure on August 25 had a very distinct bottom mixed layer (15–20-m depth, defined using a temperature change threshold of $\Delta T < 0.01\,°C$), the salinity profile was constructed to have uniform salinity at this depth interval based on the bottom measurement and through linear interpolation for the remainder of the water column. In case B2, the depth-average of the interpolated temperature (black dashed line in Fig. 4) and salinity were used as initial conditions to generate a uniform water column (i.e., mixed initial profile). Note that case B2 does not include the full effects of storm mixing and the initial total heat content in the water column was the same between cases B1 and B2. As such, another experiment (case B3) was conducted with the same conditions as case B2 except with a generic storm heat loss based on an open ocean category 3 storm following ref. [9], so the initial uniform temperature profile was cooler than that in case B2 by 0.6 °C. In all three cases, the salinity evolved as determined by the vertical mixing in the model, however, this was only relevant to case B1 since cases B2 and B3 were initialized with uniform depth-average profiles (i.e., fully mixed).

Another eight numerical experiments (cases C's in Table 1) were conducted to examine the depth dependency of the compounding impacts using the idealized initial water column thermal profiles based on more general conditions, roughly guided by the conditions in the northern Gulf of Mexico[9]. These eight paired cases consisted of paired runs for four different water depths (20, 30, 40, and 60 m) with and without initial thermal stratification at the onset of the model run. In the stratified cases (cases C1a–C4a), the thermal profiles consisted of a 6-m surface mixed layer and a bottom boundary layer over the bottom 25% of the water column, separated by a thermocline with a temperature change of 0.3 °C m$^{-1}$ (solid profiles in Fig. 4). This thermocline value represents an intermediate rate of change on the shelf in the eastern Gulf of Mexico: ~0.6 °C m$^{-1}$ at site CP and ~0.1 °C m$^{-1}$ for the west Florida shelf[9]. In the mixed cases (cases C1b–4b), the profiles associated with the stratified cases were depth-averaged to produce a uniform initial thermal structure (dashed profiles in Fig. 4). These eight idealized experiments used a uniform salinity profile to avoid additional complications from salinity stratification. All the C cases were run in a similar manner to the B cases beginning just after TS Gordon (September 7) and ending just before Hurricane Michael (October 7). For these depth paired model runs (e.g., cases C1a and C1b), the difference in the depth-average temperature was determined over the model period to assess the potential impact of a mixing event over the study period for different water depths. An additional four cases with a generic storm heat loss (similar to case B3) were conducted and compared to cases C1b–C4b, but the resulting depth-average temperatures were generally similar, only slightly lower and thus were omitted for brevity.

## Data availability
Data are publicly available from various sources stated in the "Methods" section.

## Code availability
The codes used in this study are available upon request to the corresponding author. The references in the "Methods" section provide specific details of the calculations in the codes.

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

## Acknowledgements

This work would not have been possible without the help of the Tech Support Group at the Dauphin Island Sea Lab. A portion of this work was conducted at the Jet Propulsion Laboratory, California Institute of Technology, under contract with NASA. This research was made possible by the NOAA RESTORE Science Program (NA17NOS4510101 and NA19NOS4510194) and NOAA NGI NMFS Regional Collaboration Network (18-NGI3-61).

## Author contributions

B.D. contributed to conceptualization, investigation, formal analysis, and writing (original draft). J.C. contributed to data curation and formal analysis. S.F. contributed to investigation and formal analysis. G.L. and K.P. contributed to data curation and investigation. T.L. and the other authors contributed to the writing (review and editing).

## Competing interests

The authors declare no competing interests.
