## [Peer Review File · Nature Communications]

REVIEWER COMMENTS

Reviewer #1 (Remarks to the Author):

The manuscript "Hurricanes and heatwaves: Compounding processes drive extreme events in the coastal ocean" by Dzwonkowski et al. describes an interesting scenario in which atmospheric and oceanic conditions and events contributed to extreme ocean temperatures. The study of compounding processes for extreme events and specifically marine heatwaves is highly topical and I expect that it would interest many readers. The motivation for these kinds of studies is explained clearly in the introduction and the first section in the result.

The authors argue that a tropical storm led to well-mixed ocean conditions in the coastal region, followed by an atmospheric heatwave, causing extreme ocean heat content. The enhanced ocean temperatures may have contributed to intensification of Hurricane Michael. While there are three-dimensional factors, such as advection, contributing to the temperature structure, they focus on the period of near-surface warming with a one-dimensional model. The model is used to show that the heat content is greater with an atmospheric heatwave following a mixing event (from the tropical storm) than if the waters were initially stratified. They suggest that these processes are applicable to a greater region than the mooring location.

My main comments are there are some areas of the manuscript where mechanisms are more inferred than demonstrated through observations (such as the role of ocean current temperature advection and the spatial footprint of wind stress impacts over the shelf). While the ocean heat content was higher with an atmospheric heatwave following a mixing event compared with initially stratified waters, the surface temperature is greater with the stratified case. At the air-sea interface, I would expect that the sea surface temperatures are important for feedback to the atmosphere. The manuscript and material are conceptually attractive for a reader and I think that the presentation could be improved to help support the main points, as I am currently wanting to know more information. My main comments are in regard to content structure, figure presentation, and methodology.

Major comments.

Line 73. Overall the paper is well organised. However, the content in this section on the 'Extreme nature of Hurricane Michael' is not part of the results and report information that is more along the lines of introductory material. This section provides more detailed information on the hurricane and background setting but does not clearly demonstrate results.

Line 321. The methods for derived quantities would benefit from further explanation. How does the identification of an extreme events (over a 2-day threshold) compare when a 5-day threshold is used as in the methodology presented in Hobday et al. 2016? In addition, does the method mean that all values over 2-days exceeded the 2 standard deviations? Diurnal warming may cause a temporally short but extreme value and could affect the identification of marine heatwaves through the proposed method.

In addition, how is the climatological mean and standard deviation at each time calculated in this study? The plots appear to indicate that the climatological mean and standard deviation include the diurnal and tidal component. The methods used here for identifying periods of extremely warm temperatures appear to differ than Hobday et al. 2016, which considers daily temperature values. In Figure 2, I don't understand why the 2 standard deviation exceeds or borders the maximum observed value; I am assuming that the maximum observed value is from the entire time series for that time of the year?

Minor comments.

Line 87. The sentence ending here regarding costs and fatalities needs a reference.

Line 105. What period of time was used to evaluate that this period was the warmest of the 13 years? The statement that it was the "warmest year on record" might need to be rephrased given that longer remotely sensed SST data could be used evaluate the temperatures.

Line 111-113. The onshore Ekman transport, downwelling, and subsurface warming has been inferred based on the mooring temperature profile. Are there ocean velocities available from the mooring that can be used to demonstrate this process? Or could the surface SST evolution show the progression of warmer water onshore?

Line 188-190, Figure 3. The one-dimensional model has been used to show that the ocean heat content is greater in B2 (black, initially well-mixed) and B3 (blue, initially well-mixed and with heat loss) than in B1 (red, initially stratified). However, in regard to near-surface ocean temperatures and feedback to the atmosphere, isn't the sea surface temperature extremes potentially of more importance here? In that case, Figure S4(a) (B1; stratified) actually shows that the surface temperatures and depth-average temperatures over the upper 10 m are greater than in B2 (initially well-mixed).

Line 318. Please include the published references used in the formulas listed on the website, since personal website pages are not necessarily permanent or archived records. Another option is to include the formulas here with references.

Line 335. When did the authors access the NOAA OISST dataset as listed here?

<https://www.ncdc.noaa.gov/oisst> According to the website, there has been an update in the dataset (v2.1) available since March 2020. This update is an improvement on the previously available dataset especially for correcting satellite SST biases. The authors may want to check that their results are consistent with version 2.1.

Figures.

Figure 1 caption, Line 537-538. Is the SSTA shown here equal to (mean SST – climatological mean SST – 2 standard deviations)? If so, I find that confusing and it would mean that there are regions of negative values plotted that are actually positive SST anomalies. Why not show the SSTA as (mean SST – climatological mean SST) and then indicate, using contours for example, areas that exceed 2 standard deviations?

Figure 1 and Figure 2. For the dates in the plot, I would recommend labelling them as "5 Sep" or "Sep 5" instead of "09/05", given that some readers might expect that the day is included prior to the month. Similarly, please see other areas where dates are included, such as Line 549.

Figure 2. The subplots are missing boundaries at the outer edges of the plots. In particular, subplots (a) and (d) would benefit from the upper boundary and include y-tick values for 32 deg C (as in (b)). The subplots would benefit from y-tick marks pointing outward, as in (b). In the caption, please include a reference for the direction angle, i.e. what direction is north. On Line 550, the line is thin rather than thick. In (c), there appears to be warming at approximately 10 m depth and it is unclear why that occurs. Is that warm feature caused by advection and density compensating?

Figure 3. In (b) and (c), label the y-axes with T_{da} as in Figure 2(a). Consider including a small legend in the available space adjacent to (c) and (d), which has information on each of the cases (e.g. mixed, stratified) to guide the reader. For the different cases, consider plotting the Case A curves with different colours from Case B or C. In (b) and (c), consider using the same scale for the y-axis from 24 to 30°C for comparison. I actually found Figure S4 easier to interpret than Figure 3(c).

Figure S5. Define TD and TS in the caption. Possibly TD's location is not needed to include?

Figures, general comment. There are a number of figures placed in the supplementary material, which adds more effort for the reader to go back and forth between documents. If supplementary figure or material is important for the main results, I would recommend presenting it in the main manuscript. For example, on Line 200, the spatial extent of mixing is discussed but the wind field is not explicitly shown in the figures, for example the mean wind stress fields during main periods of interest. Currently, Figure S5 is referred to for the spatial footprint but is located in the supplementary material.

Typos.

Line 101. Is "late October" supposed to be "late September"?

Line 218. Replace "associated" with "associated with".

Line 327. Replace "on shelf" with "on the shelf".

Line 401 and Figure S3. Replace "TOGO" with "TOGA".

Line 545. Replace "times series" with "time series".

Reviewer #2 (Remarks to the Author):

The current study is a novel research on compound events and marine heatwaves. Currently, there is not similar work in the literature, (to my knowledge) where the 3D heat content of a marine heatwave is looked in relation to the development of a hurricane. Therefore, the legitimacy of its publication is clear, since more extreme warm events in the ocean and intense hurricanes have been seen to occur in the last decade (or so), globally, and are expected to continue to occur in the future. The inteconnection between the hurricanes, which are strongly "fuelled" by warm ocean temperatures, and the mixing of surface heat anomalies down to the water column, which are conducive to the development of marine heatwaves, is therefore of imminent interest to the scientific community, in the context of climate change.

I recommend the publication of this study, after certain points in the manuscript have been addressed, however. To this respect, I attach a document with all my comments on the manuscript.

Review: Title:

Hurricanes and heatwaves: Compounding processes drive extreme events in the coastal ocean

Major Remarks:

How do the writers explain the “lingering” of heat anomalies at depth in the area of interest, in terms of the local oceanographic features? There were no advective processes in the continental shelf, which could have (or not) resulted in the ventilation/cooling of deeper layers? At the moment, the study explains how the surface heat anomalies were transferred to the bottom. It would be useful to make an estimation as to why these anomalies however, remained present at depth. Was it because the local circulation was somehow “inactive” or there is no circulation in the area that could cool down the water column? This could be mentioned in the discussion section.

Line 159: Why is the case A1 the chosen scenario? A2 in Supplementary material also looks similar to the post Gordon situation and also the timeseries of A2 in Fig.3c were closer to the observations than the A1 .How do the writers explain this?

Line 163: How is the choice of the A1 scenario indicate a control of vertical mixing apart from surface heat fluxes? Earlier in the text A1 was mentioned as a scenario testing a surface heat flux forcing.

Line 167: A somewhat 2-level stratification appears to be also present in B2 early in September (Fig. S4). Is that what the writer mean when they say “..the mixed case restratified but much more weakly due to the warm thermal conditions at depth ”? Where is the re-stratification in Fig. S4?

Line 171: Which coupled events do the writers mean here? Not clear. Re-write it or explain it better. I thought the 1°C increase stemmed from just the mixing of the water column in the scenario B2,B3.

Minor Remarks:

Line 55-58: This whole sentence could be rearranged to :

“...This critical gap in understanding is in large part due to the lack of **observatioal** data during extreme events, that **could** ~~put an event in historical context and/or~~ determine the antecedent conditions and process(es) that generating such events (4), which, however, by their definition, are rare and therefore difficult to obtain”

Line 58: “observational challenges”? Is this a legitimeate expression?

Line 95: “Potential to having contributed..”

Line 115: “Further homogenized the temperature”

Line 151: “..three-dimensional nature of “

Line 187: “Not surprisingly”

Line 206: “More importantly”

Line 218: “ .. downwelling effect associated with TS Gordon was clearly important..”

Fig3. Better to use the same colors for the equivalent experiments A1,B1...(red) and A2, B2 (blue) in Fig3c,d as in Fig3.b and keep the black for the observations for consistency.

-What is this dashed line in Fig.3a? Does it correspond to the different timing of water column homogenization captured by the model scenario? If so, better to be stated in the caption of the figure.

Modeling setup and scenarios section is not clear to me. Throughout the entire paper the description of the experiments was

somewhat vague. Therefore, this section should be re-written more clearly, and a more organized explanation/summary of the different methods followed should be given, one by one, e.g. following the structure of Table 1, which is not very informative at the moment. For example, how did the experiments happen? Changing one factor at a time and keeping the other ones constant or not? Was there a combination of A1,B1 scenario? Currently, it is not very clear for the reader what happened in the different experiments.

Response to Reviewer Comments

We thank the reviewers for the helpful comments as well as their time and effort. The reviewers' comments have been incorporated into the revised manuscript, which we believe has significantly improved the clarity of the presentation. The line numbers refer to those in the unmarked revised manuscript.

Reviewer #1

1-1) My main comments are there are some areas of the manuscript where mechanisms are more inferred than demonstrated through observations (such as the role of ocean current temperature advection and the spatial footprint of wind stress impacts over the shelf).

We have addressed this concern following your comments and suggestions below. For example, we have added a plot of current velocity from the mooring location (Fig. 2c,d) and provided relevant comments in various locations (Line 109-111). The current patterns at the mooring site are consistent with inferred mechanisms indicated in the manuscript.

1-2) While the ocean heat content was higher with an atmospheric heatwave following a mixing event compared with initially stratified waters, the surface temperature is greater with the stratified case. At the air-sea interface, I would expect that the sea surface temperatures are important for feedback to the atmosphere.

We agree with the comment that SST is critical in the feedback to the atmosphere. By focusing on the depth-average temperature, our intention was to capture the expected SST that would impact the storm system. Most tropical cyclones would be expected to mix the water column over the shallow shelf prior to the arrival of the core of the hurricane, i.e. the region responsive to the SST (Price 2009). This would be the case at the mooring site that is only 20 m deep. As a result, other metrics besides SST better represent the ocean-storm interaction, e.g. a fixed depth depth-average temperature or dynamic depth-average temperature (Price 2009). These properties have been shown to better predict intensification when compared to SST or TCHP (e.g. Balagura et al. 2015, 2018). Thus, we focus on the depth-average temperature. Comments and references to this affect have been added to the manuscript methods (Lines 353-357).

Major comments.

1-3) Line 73. Overall the paper is well organised. However, the content in this section on the 'Extreme nature of Hurricane Michael' is not part of the results and report information that is more along the lines of introductory material. This section provides more detailed information on the hurricane and background setting but does not clearly demonstrate results.

Yes, we struggled to find the right spot for this section. Based on your comments, we have moved it out of the results and into its own background section (Lines 71-89).

1-4) Line 321. The methods for derived quantities would benefit from further explanation. How does the identification of an extreme events (over a 2-day threshold) compare when a 5-day threshold is used as in the methodology presented in Hobday et al. 2016? In addition, does the method mean that all values over 2-days exceeded the 2 standard deviations? Diurnal warming may cause a temporally short but extreme value and could affect the identification of marine heatwaves through the proposed method.

Based on these comments, we think it is better to use the conventional methods recommended by Hobday et al. 2016 in determining marine heatwaves for both the SST and depth-average temperatures. Comments to this effect have been added to the manuscript and all the related figures have been modified to reflect this change (Fig 1 and 2). (Lines 104,125, 199, 360-366, 388-390)

1-5) In addition, how is the climatological mean and standard deviation at each time calculated in this study? The plots appear to indicate that the climatological mean and standard deviation include the diurnal and tidal component. The methods used here for identifying periods of extremely warm temperatures appear to differ than Hobday et al. 2016, which considers daily temperature values. In Figure 2, I don't understand why the 2 standard deviation exceeds or borders the maximum observed value; I am assuming that the maximum observed value is from the entire time series for that time of the year?

We should have been clearer on this point. We were trying to be very concise for word count purposes, but this was at the expense of clarity. Yes, we include the diurnal and tidal components as Hobday et al. 2016 recommends using the longest and highest resolution data possible. For the hourly data, we calculate the ensemble average at each time step over the 14 year record with typically 13 hourly values, i.e. 13 fall seasons, going the calculation of the mean. The 90th percentile was determined from this same set of values at each time step. Comments to this effect have been included in the revised manuscript (Lines 363-366).

For the OISST, we use the full 38-year record to determine the climatology and 90th percentile at each pixel at each time step. These data were used to determine a spatial perspective of the marine heatwave period identified at the mooring site. As such, the period of Sep 27-Oct 6 was used as a window to determine if a marine heatwave was present or absent within this time period at each pixel. The time period covers the dates when the depth-average temperature at mooring was consistently above the 90th percentile value until 3 days prior to landfall of Hurricane Michael. Ending the averaging period 3 days prior to landfall is similar to the pre-storm SST value used in Balugura et al. (2020). These comments and clarifications have been added to the manuscript (Lines 391-400).

Minor comments.

1-6) Line 87. The sentence ending here regarding costs and fatalities needs a reference.

The reference has been added. It was from the same reference as the previous line (Line 85). Sorry for any confusion.

1-7) Line 105. What period of time was used to evaluate that this period was the warmest of the 13 years? The statement that it was the “warmest year on record” might need to be rephrased given that longer remotely sensed SST data could be used evaluate the temperatures.

This is a good point. This has been rephrased to “By comparison, the 2018 depth-average temperature was well above these conditions (at or above the 90th percentile), making it the warmest year observed in the in-situ time series at site CP and exceeding any other year by 0.5-1 °C for this late September/early October time period.” Lines 103-106

1-8) Line 111-113. The onshore Ekman transport, downwelling, and subsurface warming has been inferred based on the mooring temperature profile. Are there ocean velocities available from the mooring that can be used to demonstrate this process? Or could the surface SST evolution show the progression of warmer water onshore?

We have examined the AWAC data from the deployment period and it is consistent with the inference. We have included additional figures and texts associated with this data (Fig. 2c,d and Lines 109-111 and 367-375).

Originally, we did not include these data for two reasons. 1) Space, we were trying to be very concise in this manuscript. 2) In part because of point 1) we are planning a follow-up manuscript that was going to include this data. Regardless, we have included the current velocity data to strengthen this manuscript as we think this is a good point by the reviewer.

1-9) Line 188-190, Figure 3. The one-dimensional model has been used to show that the ocean heat content is greater in B2 (black, initially well-mixed) and B3 (blue, initially well-mixed and with heat loss) than in B1 (red, initially stratified). However, in regard to near-surface ocean temperatures and feedback to the atmosphere, isn't the sea surface temperature extremes potentially of more importance here? In that case, Figure S4(a) (B1; stratified) actually shows that the surface temperatures and depth-average temperatures over the upper 10 m are greater than in B2 (initially well-mixed).

See Comment 1-2)

1-10) Line 318. Please include the published references used in the formulas listed on the website, since personal website pages are not necessarily permanent or archived records. Another option is to include the formulas here with references.

The references have been added (Line 360).

1-11) Line 335. When did the authors access the NOAA OISST dataset as listed here? <https://www.ncdc.noaa.gov/oisst> According to the website, there has been an update in the dataset (v2.1) available since March 2020. This update is an improvement on the previously available dataset especially for correcting satellite SST biases. The authors may want to check that their results are consistent with version 2.1.

Thank you for this point. We were using the older version of this data. We have downloaded the new data and the results are presented in the revised Fig. 1. A clarifying comment has been added to the Methods section (Lines 385-388, 397-400).

1-12) Figure 1 caption, Line 537-538. Is the SSTA shown here equal to (mean SST – climatological mean SST – 2 standard deviations)? If so, I find that confusing and it would mean that there are regions of negative values plotted that are actually positive SST anomalies. Why not show the SSTA as (mean SST – climatological mean SST) and then indicate, using contours for example, areas that exceed 2 standard deviations?

The figure has been revised as suggested. We have now plotted the SSTA with the contour for the absence or presence of a marine heatwave prior to the arrival of Hurricane Michael. See revised Fig. 1 and Lines 391-400 and 665-669.

1-13) Figure 1 and Figure 2. For the dates in the plot, I would recommend labelling them as “5 Sep” or “Sep 5” instead of “09/05”, given that some readers might expect that the day is included prior to the month. Similarly, please see other areas where dates are included, such as Line 549.

The revisions have been made in all figures.

1-14) Figure 2. The subplots are missing boundaries at the outer edges of the plots. In particular, subplots (a) and (d) would benefit from the upper boundary and include y-tick values for 32 deg C (as in (b)). The subplots would benefit from y-tick marks pointing outward, as in (b). In the caption, please include a reference for the direction angle, i.e. what direction is north. On Line 550, the line is thin rather than thick.

While we made most of these changes, the other changes in the plot (i.e. addition of the 90th percentile and removal of the max and 2*std band) have hopefully addressed concerns with the visual appearance of this plot. See Revised Fig. 2 and Fig. S1.

1-15) In (c), there appears to be warming at approximately 10 m depth and it is unclear why that occurs. Is that warm feature caused by advection and density compensating?

Yes, this is an interesting feature. From the available data, it is not clear what is causing the warming in this region of the water column, however we have provide comments about possible causes based on your suggestion.

“Interestingly, there was a mid-water column warming in late September (~28th-29th, Fig. 2e). From the available data it is difficult to determine the cause of this event, however the warming may have been generated by advection or density compensation.” Lines 129-132

1-16) Figure 3. In (b) and (c), label the y-axes with T_{da} as in Figure 2(a). Consider including a small legend in the available space adjacent to (c) and (d), which has information on each of the cases (e.g. mixed, stratified) to guide the reader. For the different cases, consider plotting the Case A curves with different colours from Case B or C. In (b) and (c), consider using the same scale for the y-axis from 24 to 30°C for comparison. I actually found Figure S4 easier to interpret than Figure 3(c).

All of these recommendations have been included in the revised Figures.

1-17) Figure S5. Define TD and TS in the caption. Possibly TD's location is not needed to include?

These terms have been defined in the caption of revised Fig 6.

1-18) Figures, general comment. There are a number of figures placed in the supplementary material, which adds more effort for the reader to go back and forth between documents. If supplementary figure or material is important for the main results, I would recommend presenting it in the main manuscript. For example, on Line 200, the spatial extent of mixing is discussed but the wind field is not explicitly shown in the figures, for example the mean wind stress fields during main periods of interest. Currently, Figure S5 is referred to for the spatial footprint but is located in the supplementary material.

This is great suggestion and we have moved all but one of the figures from the supplemental materials into the manuscript. As a part of this change, we have reorganized the figures in the manuscript accordingly (See revised Fig 4-6).

1-19: Typos

Line 101. Is "late October" supposed to be "late September"?

Line 218. Replace "associated" with "associated with".

Line 327. Replace "on shelf" with "on the shelf".

Line 401 and Figure S3. Replace "TOGO" with "TOGA".

Line 545. Replace "times series" with "time series".

Thank you of these edits. All the typos identified by the reviewers have been addressed.

Reviewer #2

Major Remarks:

2.1) How do the writers explain the “lingering” of heat anomalies at depth in the area of interest, in terms of the local oceanographic features? There were no advective processes in the continental shelf, which could have (or not) resulted in the ventilation/cooling of deeper layers? At the moment, the study explains how the surface heat anomalies were transferred to the bottom. It would be useful to make an estimation as to why these anomalies however, remained present at depth. Was it because the local circulation was somehow “inactive” or there is no circulation in the area that could cool down the water column? This could be mentioned in the discussion section.

This is a good point. Yes, velocity data from site CP suggest the circulation was weak post TS Gordon. We have added a paragraph on this topic in Discussion. See Fig. 2c,d and Lines 264-275

2-2) Line 159: Why is the case A1 the chosen scenario? A2 in Supplementary material also looks similar to the post Gordon situation and also the timeseries of A2 in Fig.3c were closer to the observations than the A1 .How do the writers explain this?

Yes, we should have been clearer on this point. We have added additional material to the manuscript on this point: “While case A2 had SST and depth-average temperature closer to the observed conditions, the heat flux parameterizations used in case A1 was selected for the subsequent model experiments for two reasons. ... The relatively high correlation between case A1 and observations ($r=0.89$) indicates that the post-storm thermal structure of the water column was primarily a one-dimensional balance driven by surface heat fluxes and vertical mixing. Thus, this simple one-dimensional model (case A1) was used in the subsequent model experiments to further examine the thermal structure with and without compounding processes.” (Lines 168-182)

2-3) Line 163: How is the choice of the A1 scenario indicate a control of vertical mixing apart from surface heat fluxes? Earlier in the text A1 was mentioned as a scenario testing a surface heat flux forcing.

Our writing was unclear. We have revised this section of the text (See Comment 2-2 and/or Lines 177-182)

2-4) Line 167: A somewhat 2-level stratification appears to be also present in B2 early in September (Fig. S4). Is that what the writer mean when they say “..the mixed case restratified but much more weakly due to the warm thermal conditions at depth ”? Where is the re-stratification in Fig. S4?

Yes, we tried to clarify this point. This has been revised (Lines 186-192).

2-5) Line 171: Which coupled events do the writers mean here? Not clear. Re-write it or explain it better. I thought the 1°C increase stemmed from just the mixing of the water column in the scenario B2, B3.

We meant the passage of TS Gordon followed by the atmospheric heatwave. The text has been revised. (Lines 194-199)

Minor Remarks

2-6) Line 55-58: This whole sentence could be rearranged to : “...This critical gap in understanding is in large part due to the lack of **observational** data during extreme events, that **could** put an event in historical context and/or determine the antecedent conditions and process(es) that generating such events (4), which, however, by their definition, are rare and therefore difficult to obtain”

This has been revised based on this comment: “This critical gap in understanding is in large part due to the lack of observational data during extreme events, which by their definition are rare. Such data could put events in historical context and/or determine the antecedent conditions and process(es) that generate such events (4).” Lines 55-58

2-7) Typos and grammatical corrections

Line 58: “observational challenges”? Is this a legitimeate expression?

Line 95: “Potential to having contributed..”

Line 115: “Further homogenized the temperature”

Line 151: “..three-dimensional nature of “

Line 187: “Not surprisingly”

Line 206: “More importantly”

Line 218: “ .. downwelling effect associated with TS Gordon was clearly important..”

Thank you. These have all been modified as suggested.

2-8) Fig3. Better to use the same colors for the equivalent experiments A1,B1...(red) and A2, B2 (blue) in Fig3c,d as in Fig3.b and keep the black for the observations for consistency.

-What is this dashed line in Fig.3a? Does it correspond to the different timing of water column homogenization captured by the model scenario? If so, better to be stated in the caption of the figure.

The figures have been re-organized as well as revised as suggested. The vertical dashed line represents the start time of the 2nd and 3rd sets of model experiments (i.e. after the passage of TS Gordon). This is now stated in the caption of revised Fig. 3.

2-9) Modeling setup and scenarios section is not clear to me. Throughout the entire paper the description of the experiments was somewhat vague. Therefore, this section should be re-written more clearly, and a more organized explanation/summary of the different methods followed should be given, one by one, e.g. following the structure of Table 1, which is not very informative at the moment. For example, how did the experiments happen? Changing one factor

at a time and keeping the other ones constant or not? Was there a combination of A1,B1 scenario? Currently, it is not very clear for the reader what happened in the different experiments.

This is a very good point. Our effort to be very concise resulted in a lack of clarity. The model result (**Compounding impacts on heat content** Section) and methods (**Modeling setup and scenarios** Section) sections have been revised to more clearly explain each scenario, the rationales, and the way through which the model experiments were conducted.

REVIEWERS' COMMENTS:

Reviewer #1 (Remarks to the Author):

I thank the authors for their considerations and efforts to address my first review comments. The manuscript has greatly improved in clarity.

I noticed a couple minor details that I did not see initially, or they were included in the latest revision, and there may be other minor details that I did not find. I am fine with these details to be addressed a later stage and I do not need to see another revision. This manuscript is an important study for understanding ocean heat extremes in the coastal environment resulting from compound extreme events.

Minor comments.

Line 30. This line is missing a comma and noun. Replace "trends can" with "trends, they".

Line 49. For "suggested/shown", I would choose one or the other and remove the slash or write "suggested or shown".

Lin 62-63. Please include commas after "content" and after "landfall".

Line 132. Replace "overall all" with "overall".

Line 156. Replace "Figs." with "Fig.".

Line 248. Replace "generated mixing impacted" with "generated mixing that impacted".

Line 256. I would consider removing the phrase "(if any)".

Line 271. Replace "showed of a" with "showed a".

Line 309. I think you mean "significant decrease" rather than "significant increase".

Line 645. The name of one author is underlined when it probably is not meant to be underlined.

Reviewer #2 (Remarks to the Author):

All my comments about the current version of the manuscript are included in the uploaded/attached document below.

Attachment:

2nd Review: Title:

Hurricanes and heatwaves: Compounding processes drive extreme events in the coastal ocean

Revised manuscript:

The writers have addressed all the previous reviewer's comments and now the manuscript has a more clear presentation of methods and results. However, there are still a few remaining points that need to be improved, as they are currently not very clear in the way they are written. I advise the publication of this study after the final comments have been addressed. See my comments below:

Major Comments:

Line 196-199: Where do the writers demonstrate that nearly 50% of the observed deviation from the long-term mean state is due to to this ~ 1 ° C increase in sea water temperature coupled with the subsequent atmospheric heatwave-driven temperature increase?

Line 199: Where do the writer demonstrate that if it wasn't for this ~ 1 ° C effect, then it would have been an otherwise above average event? What if there was an increase of temperature higher than the climatology but still lower than 1 ° C?

Line 200-218: This paragraph related to Fig.4 tries to convey one of the most important messages of this study. However, currently it is not yet clear to me what do the writers mean in their description. Some more clarifications are still needed in some points. See some of my comments below:

- **Line 208:** Do you mean that the stratified and mixed initial conditions resulted in small differences in the depth-averaged temperature? What do you mean by "*showed very little effect*"?

- **Line 209.** By SST ("*..larger differences in the SSTs*") do you mean the depth-average temperature of the water column? In a well-mixed situation, the SST represents more or less the temperature of the mixed layer depth (in your case each of the vertical limits you have chosen in each experiment. In a stratified case though the average temperature of the water column selected is not represented by SST.

Maybe you could change SST into temperature, unless you mean something else here. It is not clear currently.

Line 211-218: If I understood well so far, there was little change/difference between the depth-average temperature (heat content) of the 20m water column experiments (mixed and stratified).Then, between 20m-30m there was a considerable increase of the heat content when there was an event coupling. However, at depth below 40m depth it seems that the heat content increase does not happen, or is not considerable. So indeed **Line 215** argument is true: "*.. changes in depth and hydrographic structure will*

affect the extent to which compounding processes intensify the warming of the water column". However, in **Line 218**: "*.. the coupling of a mixing event followed by an atmospheric heatwave does enhance heat content of the water column relative to an atmospheric heatwave without a mixing event...*". Is that true for all the depths, even below 40m? Or the considerable increase of heat content happens only when mixing happens up to the first 20-30m? Since in **Line 212** the writers state that: "*... However, this increasing heat content difference with water depth will eventually be mitigated by the depth-average temperature dropping to or below 26 ° C (e.g., gray dashed line in Fig. 4)*". Are the writers sure that they can generalize their hypothesis for every mixing depth prior to an atmospheric heatwave? Unless, the writers mean something else here.

Minor Comments:

Line 20-22: Maybe move the "*..a critical contributor to storm intensity..*" to the end of the sentence. It makes more sense to declare first your results, showing the contribution of extreme heat to the intensification of Hurricane Michael, and then generalize that this could be a generally important contributor to storm intensity.

Line 29: "*devastating in and of themselves*"? What do the writers mean by this? Maybe they meant "*..devastating for them*"?

Line 43: Maybe change “Importantly” to “More specifically”?

Line 101: In which years does the “*long-term mean*” exactly refer to? What is the exact period?

Line 127: What do the writers mean by a “*slower..event compared to the TS Gordon*”? How can a marine heatwave be a “slower” event compared to a storm?

Line 132: Typo. “The overall all”

Line 143: Missing. “As a function of wind...”

Line 427: Maybe better to write “ as Hurricane Michael approached”?

Line 171: Typo.” A priori”

Line 202: Is the relative impact increasing with depth or with the distance offshore? Because there could perhaps be a different impact with depth but also close to the shore, no?

Line 206: Do you mean that the difference between the mixed and the stratified case is on the average depth-temperature and not on the surface SST (depth 0)? Unless by SST you mean the first few meters of the water column. If you have a stratified column the SST is different from the SST in a mixed water column.

Line 211: Do you mean that the depth-average temperature in the experiments at 30m and 40m have a difference of >0.8 C? Or between 30m and 60m? Not clear.

Line 215: Affect instead of “effect”

Line 278: Why the hurricane intensity is reduced when there is less stratification? From this study I understood that de-stratification before the passage of

a hurricane increases the heat content which can be later used to increase the intensity of the storm.

Figure 2. An option would be to put the dates in all plots (Fig.2a-f) just like in 2g. You currently show 7 plots and the reader has to go up and down to check the dates every time.

Figure. 4 : This figure is nice but needs improvement.

- Maybe change the “synthetic” to “modelled”?
- I am missing the dashed lines of the 30m and 40m run. Therefore I cannot see what is described in **Line 211** of the main text.
- Do the colored vertical dashed lines represent the vertically-averaged temperature in the idealized mixed conditions? What do the writers mean when they say that it represents “*the mixed initial conditions* (What do you mean by initial conditions?)? And if the color dashed lines represent the vertically-averaged temperature in idealized mixed conditions, shouldn't there be another line that represents the vertically-averaged temperature of the idealized stratified conditions?(In order to be able to compare the average temperature in the water column in mixed and stratified cases)

Table 1:

- Maybe you could add a column where the depth-averaged temperatures are given in each of the experiments performed. That way the reader is better informed about the differences rather than trying to assess them from Figure 4.

Response to Reviewers' Comments

We thank the reviewers for the helpful comments as well as their time and effort. The reviewers' comments have been incorporated into the revised manuscript, which we believe has significantly improved in terms of the clarity of the presentation. The line numbers refer to those in the unmarked revised manuscript.

Reviewer #1:

Minor comments.

1-1) Line 30. This line is missing a comma and noun. Replace “trends can” with “trends, they”.

This has been revised as requested

1-2) Line 49. For “suggested/shown”, I would choose one or the other and remove the slash or write “suggested or shown”.

We have added an ‘or’ as requested.

1-3) Lin 62-63. Please include commas after “content” and after “landfall”.

This has been revised as requested.

1-4) Line 132. Replace “overall all” with “overall”.

This has been revised as requested.

1-5) Line 156. Replace “Figs.” with “Fig.”.

This has been revised as requested.

1-6) Line 248. Replace “generated mixing impacted” with “generated mixing that impacted”.

This has been revised as requested.

1-7) Line 256. I would consider removing the phrase “(if any)”.

This has been revised as requested.

1-8) Line 271. Replace “showed of a” with “showed a”.

This has been revised as requested.

1-9) Line 309. I think you mean “significant decrease” rather than “significant increase”.

Thank you for highlighting this. Yes, that is correct, but in re-reading that sentence, we decided to revise this phrase to ‘...because of the significance of the water column temperature structure at depth.’ to better express that the water column structure is what is most important (Line 318).

1-10) Line 645. The name of one author is underlined when it probably is not meant to be underlined.

This has been revised as noted.

Reviewer #2

Major Comments:

2-1) Line 196-199: Where do the writers demonstrate that nearly 50% of the observed deviation from the longterm mean state is due to this ~ 1 ° C increase in sea water temperature coupled with the subsequent atmospheric heatwave-driven temperature increase?

The observed depth-average temperature was $1.7-1.9^{\circ}\text{C}$ higher than the long-term mean (Fig. 2a). The modeling analysis (cases B1-B3) indicates that the sequence of mixing followed by a subsequent atmospheric heatwave results in an $0.9-1.1^{\circ}\text{C}$ increase in the depth-average temperature. From these, we concluded that nearly 50% of the observed deviation from the long term mean state was caused by the compounding processes. We have clarified this by modifying the text to read as follows: “Given that the observed depth-average temperature was nearly 2°C above the long-term mean in early October, this ~ 1 ° C increase from the mixing event coupled with a subsequent atmospheric heatwave accounts for nearly 50% of the observed deviation from the long-term mean state (Fig. 2a) and represents a temperature change large enough to significantly impact storm intensity.” (Lines 193-197)

2-2) Line 199: Where do the writer demonstrate that if it wasn’t for this ~ 1 ° C effect, then it would have been an otherwise above average event? What if there was an increase of temperature higher than the climatology but still lower than 1 ° C?

Similar to Comment (2-1), we determined this from Fig. 2a. With an $\sim 1^{\circ}\text{C}$ contribution from the compounding processes, the effect of removing this $\sim 1^{\circ}\text{C}$ would put the depth-average temperature below the 90% percentile for a marine heatwave, but still above the long-term mean (i.e. above average event). We have clarified this by modifying the text to read as follows: “Importantly, removing this added 1°C effect from the observed depth-average temperature in early October would place the depth-average temperature well below the 90th percentile threshold associated with marine heatwaves (Fig. 2a). Thus, the compounding process observed in September of 2018, adding 1°C depth-average temperature to the water column, made what would have been an otherwise above average event into an extreme event (above the 90th percentile threshold, Fig. 2a).” (Lines 198-200)

2-3) Line 208: Do you mean that the stratified and mixed initial conditions resulted in small differences in the depth-averaged temperature? What do you mean by “showed very little effect”?

Yes. We revised the sentence to: “...showed very little effect on the overall heat content of the water column, i.e., very little difference in the depth-average temperatures (Fig. 5d).” (Lines 212-213)

2-4) Line 209. By SST (“..larger differences in the SSTs”) do you mean the depth-average temperature of the water column? In a well-mixed situation, the SST represents more or less the temperature of the mixed layer depth (in your case each of the vertical limits you have chosen in each experiment. In a stratified case though the average temperature of the water column selected is not represented by SST. Maybe you could change SST into temperature, unless you mean something else here. It is not clear currently.

You make a good suggestion. We have changed SST to ‘upper ocean temperatures’. (Line 210)

2-5) Line 211-218: If I understood well so far, there was little change/difference between the depth-average temperature (heat content) of the 20m water column experiments (mixed and stratified). Then, between 20m-30m there was a considerable increase of the heat content when there was an event coupling. However, at depth below 40m depth it seems that the heat content increase does not happen, or is not considerable. So indeed Line 215 argument is true: “.. changes in depth and hydrographic structure will affect the extent to which compounding processes intensify the warming of the water column”. However, in Line 218: “.. the coupling of a mixing event followed by an atmospheric heatwave does enhance heat content of the water column relative to an atmospheric heatwave without a mixing event...”. Is that true for all the depths, even below 40m? Or the considerable increase of heat content happens only when mixing happens up to the first 20-30m?

Yes. This is generally the correct interpretation (see Fig. 5d). At depths 40 m and deeper, the difference between the mixed and stratified cases is notable, with the ‘mixed’ model runs always resulting in warmer water column than the ‘stratified’ model runs. However, the absolute depth-average temperature begins to decrease to the point where the depth-average temperature of the 60 m case is less than 26°C (i.e. not favorable for storm intensification). We clarified the text to “In the two deepest cases (i.e., 40 and 60 m), the event coupling provided differences >0.8°C in the depth-average temperatures between the mixed and stratified cases (Fig. 4d). In addition, these modeling scenarios suggest that the increasing heat content difference with water depth will eventually be mitigated by the depth-average temperature dropping to or below 26°C (e.g., gray dashed line in Fig. 4), leading to conditions that would not be expected to favor hurricane intensification.” (Lines 215-220)

2-6) Since in Line 212 the writers state that: “ ... However, this increasing heat content difference with water depth will eventually be mitigated by the depth-average temperature dropping to or below 26 ° C (e.g., gray dashed line in Fig. 4)”. Are the writers sure that they can generalize their hypothesis for every mixing depth prior to an atmospheric heatwave? Unless, the writers mean something else here.

We did not mean to presume that we can generalize our hypothesis for every mixing depth prior to an atmospheric heatwave. We have revised this statement to be more constrained and clearer. It now reads: “In addition, these modeling scenarios suggest that the increasing heat content difference with water depth will eventually be mitigated by the depth-average temperature dropping to or below 26°C (e.g., gray dashed line in Fig. 4), leading to conditions that would not be expected to favor hurricane intensification. While additional observational and modeling work is needed to fully understand relationships between mixing and reheating events on continental shelves, our findings do indicate that changes...” (Lines 217-222)

Minor Comments:

2-7) Line 20-22: Maybe move the “..a critical contributor to storm intensity..” to the end of the sentence. It makes more sense to declare first your results, showing the contribution of extreme heat to the intensification of Hurricane Michael, and then generalize that this could be a generally important contributor to storm intensity.

This has been revised as suggested.

2-8) Line 29: “devastating in and of themselves”? What do the writers mean by this? Maybe they meant “..devastating for them”?

We agree this wording was clumsy. We revised it to: “Such events alone can be devastating and ...” (Lines 29-30)

2-9) Line 43: Maybe change “Importantly” to “More specifically”?

We have changed this as suggested.

2-10) Line 101: In which years does the “long-term mean” exactly refer to? What is the exact period?

As requested, this has been changed to “The long-term (2005-2018) mean in late September...”

2-11) Line 127: What do the writers mean by a “slower..event compared to the TS Gordon”? How can a marine heatwave be a “slower” event compared to a storm?

We agree this wording is a bit clumsy and has been removed. We just meant that the rate of warming was slower, but this not an essential point. The sentence now reads “This atmospheric heatwave produced a warming event of longer duration (relative to TS Gordon), ...” (Lines 125-126)

2-12) Line 132: Typo. “The overall all” Line 427: Maybe better to write “ as Hurricane Michael approached”? Line 171: Typo.” A priori”

This has been corrected as suggested.

2-13) Line 143: Missing. “As a function of wind...”

This sentence does refer to the role of wind forcing in the model as expressed in later part of the sentence. It was not clear to us why the reviewer feels we should start the sentence with this text, so we prefer not to make this addition at this location.

2-14) Line 202: Is the relative impact increasing with depth or with the distance offshore? Because there could perhaps be a different impact with depth but also close to the shore, no?

This is a good point. We have focused only on the depth dependence, so we have removed the comment related to distance offshore.

2-15) Line 206: Do you mean that the difference between the mixed and the stratified case is on the average depth-temperature and not on the surface SST (depth 0)? Unless by SST you mean the first few meters of the water column. If you have a stratified column the SST is different from the SST in a mixed water column.

Yes, we should have been clearer. We have changed SST to the upper ocean temperatures.

2-16) Line 211: Do you mean that the depth-average temperature in the experiments at 30m and 40m have a difference of >0.8 C? Or between 30m and 60m? Not clear.

Yes, this was poorly worded. We are referring to difference between the stratified and mixed conditions for the 40m and 60m depth cases. We have revised the sentence to state this more clearly: “In the two deepest cases (i.e., 40 and 60 m), the event coupling provided differences $>0.8^{\circ}\text{C}$ in the depth-average temperatures between the mixed and stratified cases (Fig. 4d).” (Line 215-217)

2-17) Line 215: Affect instead of “effect”

Corrected.

2-18) Line 278: Why the hurricane intensity is reduced when there is less stratification? From this study I understood that de-stratification before the passage of a hurricane increases the heat content which can be later used to increase the intensity of the storm.

Yes. We agree with the reviewer. The stratification breakdown we were referring to is associated with the immediate passage of a storm that injects colder bottom waters into the upper water column. We have clarified this point in the sentence. “... has focused on intensity reduction through stratification breakdown and the resulting injection of colder bottom water into the upper water column as storms pass over these regions.” (Lines 285-287)

2-19) Figure 2. An option would be to put the dates in all plots (Fig.2a-f) just like in 2g. You currently show 7 plots and the reader has to go up and down to check the dates every time.

We added dates at the top of the panel Fig. 2a so that the readers would see an x-axis label at the top and bottom of the figure. We hope that balances the need for better time reference points while reducing repeated text on a busy plot. We are happy to put date on the subpanels should the reviewer still prefer that. (revised Fig. 2)

2-20) Figure. 4 : This figure is nice but needs improvement. - Maybe change the “synthetic” to “modelled”? - I am missing the dashed lines of the 30m and 40m run. Therefore I cannot see what is described in Line 211 of the main text. –

We have made these changes. Our intent was to limit the number of vertical lines so as not to overcrowd the plot, however it is clear we should have presented all the depth-average lines in this figure. (revised Fig. 4)

2-21) Do the colored vertical dashed lines represent the vertically-averaged temperature in the idealized mixed conditions? What do the writers mean when they say that it represents “the mixed initial conditions (What do you mean by initial conditions?)”? And if the color dashed lines represent the vertically-averaged temperature in idealized mixed conditions, shouldn’t there be another line that represents the vertically averaged temperature of the idealized stratified conditions?(In order to be able to compare the average temperature in the water column in mixed and stratified cases)

We now realize the figure caption was not clearly written. The figure shows the initial thermal structure for all model scenarios. The vertical dashed lines indicate ‘mixed’ cases which were obtained by taking the depth-average temperature associated with each stratified case and projecting over the full water column. Thus, the depth-average temperature in the mixed and stratified cases is the same at the start of the model runs. Fig. 4 caption has been revised to clarify this.

2-22) Table 1: - Maybe you could add a column where the depth averaged temperatures are given in each of the experiments performed. That way the reader is better informed about the differences rather than trying to assess them from Figure 4.

This is a good suggestion, however after adding a column to the table with this information, we felt the table looked too congested. As such, we have added a footnote to the Table with this information. We hope that is a suitable way to address your suggestion.